# Target Detection-Based Control Method for Archive Management Robot

**DOI:** 10.3390/s23115343

**Published:** 2023-06-05

**Authors:** Cheng Yan, Jieqi Ren, Rui Wang, Yaowei Chen, Jie Zhang

**Affiliations:** 1College of Automation, Nanjing University of Science & Technology, Xiaolingwei Street, Nanjing 210094, China; ycyc001007@njust.edu.cn (C.Y.); 321110010236@njust.edu.cn (R.W.); 221110011239@njust.edu.cn (Y.C.); 2Second Academy of Aerospace Science and Industry, Yongding Road, Beijing 100854, China; 2008renjieqi@163.com

**Keywords:** YOLOV5, position estimation, robotic arm, unmanned archive repository

## Abstract

With increasing demand for efficient archive management, robots have been employed in paper-based archive management for large, unmanned archives. However, the reliability requirements of such systems are high due to their unmanned nature. To address this, this study proposes a paper archive access system with adaptive recognition for handling complex archive box access scenarios. The system comprises a vision component that employs the YOLOV5 algorithm to identify feature regions, sort and filter data, and to estimate the target center position, as well as a servo control component. This study proposes a servo-controlled robotic arm system with adaptive recognition for efficient paper-based archive management in unmanned archives. The vision part of the system employs the YOLOV5 algorithm to identify feature regions and to estimate the target center position, while the servo control part uses closed-loop control to adjust posture. The proposed feature region-based sorting and matching algorithm enhances accuracy and reduces the probability of shaking by 1.27% in restricted viewing scenarios. The system is a reliable and cost-effective solution for paper archive access in complex scenarios, and the integration of the proposed system with a lifting device enables the effective storage and retrieval of archive boxes of varying heights. However, further research is necessary to evaluate its scalability and generalizability. The experimental results demonstrate the effectiveness of the proposed adaptive box access system for unmanned archival storage. The system exhibits a higher storage success rate than existing commercial archival management robotic systems. The integration of the proposed system with a lifting device provides a promising solution for efficient archive management in unmanned archival storage. Future research should focus on evaluating the system’s performance and scalability.

## 1. Introduction

Paper archives have a unique advantage over electronic archives in terms of tamper resistance. Furthermore, some paper documents contain confidential information or personal data that cannot be converted into electronic files for storage. This highlights the continued significance of paper archive management. With the development of the field of archives management, many large enterprises have gradually increased their demand for archives management, which is a managerial and scientific task with extremely high management requirements for the depositing, borrowing, and removal of documents, and therefore, the management of paper archives is extremely demanding [1]. The implementation of an unmanned paper-based records management system offers an automated, standardized, and efficient solution for enterprise records management. Within this context, the development of humanoid archival capture robots has gained significant attention for its innovative nature and practical applications. The design of an archive access robot faces several challenges, such as the heterogeneity of storage environments, the variability of archive shelf arrangements, and the diversity of archive shelf types, which limit the standardization of an archive gripping system. To address this, incorporating a camera at the end of the robot arm can enhance the system’s adaptability to irregular placement scenarios.

The challenge of accurately accessing archive boxes stored in diverse shelving units has been a hindrance to the implementation of unmanned archive storage systems. The diverse shapes of archive boxes and variations in shelf structures result in variations in the final box placement, which can cause a failure in the robot’s task, even with slight deviations. To address this issue, a common approach is to use QR codes marked on the shelves or boxes to aid the robot in positioning and identification. However, this approach increases access failure rates in complex scenarios, and incurs substantial operation and maintenance costs due to the need for additional QR code affixing. In light of these challenges, this study proposes the use of machine vision for direct target identification to improve the performance of robots tasked with accessing archive boxes in archive repositories.

The vision servo system mentioned in this paper is mainly used for the access operation of archive boxes in unmanned archive storage. The main task is to calculate the transformation matrix from the camera to the end mechanism and to execute actions based on the transformation matrix.

Machine vision is a promising solution for the identification of archive boxes and shelf grid openings, allowing for successful task execution in the presence of irregular structures. While a research and development organization in Shandong, China has implemented the use of 2D codes and shelf mapping to enable target recognition, this indirect approach is limited by the requirement for consistent shelf structure. Alternative methods using machine vision may offer more robust and cost-effective solutions for the identification of archive boxes and shelf grid openings.

In Guangdong, China, an alternative approach involves the direct recognition of QR codes on archive boxes, providing spatial location without being impacted by structural inconsistencies. However, this approach requires the manual attachment of QR codes to each box and may reduce the success rate if not attached parallel to the box.

Our study suggests using machine vision for the direct recognition of archive boxes and shelf grid openings, allowing the robot to estimate the target’s position without the need for auxiliary media or pre-defined spatial mapping functions. This approach ensures successful task execution even with irregular structures of archives and shelves.

Computer vision is a technology that allows computers to process and analyze images, allowing them to recognize objects and features within these images through the use of techniques such as contouring or machine learning [2,3]. The application of computer vision in robotic systems has been demonstrated to facilitate autonomous task execution through image recognition and interpretation. Specifically, computer vision-based control systems can be utilized in robotic arm systems to perform complex operations such as sorting, filtering, picking, and placing, without human intervention. These techniques have been reported in previous studies, such as those described by Abbood et al. [4,5,6,7,8].

The complexity of the archival storage environment poses challenges for visual orientation, such as poor structural consistency of archival boxes and shelves, viewpoint obscuration, and light variations. As a result, the task of accurately detecting and estimating the position of targets in this environment remains a significant challenge in the development of archival management robotic systems. Learning-based detection algorithms have proven to be important in several disciplines, including intelligent surveillance systems [9], wireless sensors [10,11], and secure traffic systems [12]. In the last few years, convolutional neural networks (CNNs) have achieved good results in several computer vision tasks. You Only Look Once (YOLO) is an effective method for visual detection [13,14].

This work delves into the problem of target detection and position estimation for robotic arm systems in unmanned archive repositories. A computer vision-based approach is proposed to address the challenges posed by the complex environment of archive storage rooms. A 6-degrees-of-freedom robotic arm system equipped with an RGB+depth vision sensor has been designed and implemented to achieve the archive box access function. A feature area-based automatic matching vision localization algorithm is proposed to effectively handle the inconsistencies in the shapes and structures of archive boxes and grids, field of view obscuration, and lighting conditions [15,16,17,18,19,20]. The proposed approach demonstrates a promising solution for the application of robotic arm systems in unmanned archive repositories.

In this study, we propose a target location estimation and action execution control model for archive management tasks in irregular environments. The proposed model is evaluated through comparative experiments with a conventional 2D code-based system, providing insights for optimizing unmanned archive repositories. This research offers a new direction for the construction of unmanned archive repositories and contributes to the development of target location estimation algorithms for archival management tasks in challenging environments.

The contribution of this work is as follows:Applying YOLOV5’s method to the identification of archival boxes and archival shelf grids, and experimentally verifying the feasibility of identifying the marked feature areas, demonstrating that the method can be applied to the niche area of paper archival access.An algorithm for estimating target locations through the use of feature region matching and ranking is proposed to enhance the system’s ability to adapt to variations in the structures of archival shelves and boxes.By comparing with other commercial file management robots on the market, the comparison experiment results show that using the visual positioning method is faster and more accurate than the QR code positioning method, which has better practical value.

## 2. Model Construction

### 2.1. Theoretical Model

In order to achieve automatic recognition and access to archive boxes by a robot, a vision-based servo system was designed for the purpose of detecting and estimating the positions of archive boxes (or compartments within an archive shelf).

The proposed vision servo system for automatic recognition and access to archive boxes includes a camera, a robotic arm, and a main controller. The camera captures RGB images and depth information, which are processed by the main controller to estimate the location of the target archive box. The robotic arm is then controlled to reach the target position and attitude, and the end-actuator grasps or places the target object (archive box) as needed. The system is illustrated in Figure 1.

In this field of research, the application of deep learning-based detection algorithms to target detection has emerged as a prominent approach. With advancements in machine vision technology, the accuracy and processing speeds of these algorithms have undergone continuous improvement. The current state-of-the-art in target detection relies heavily on the utilization of deep learning methods. Deep learning-based target detection algorithms include faster R-CNN [21], SSD [22], YOLOV3 [14], YOLOV5, etc. The YOLOV5 method demonstrates improved robustness, generalization, and accuracy when compared to traditional target detection methods.And the schematic block diagram of YOLO algorithm is shown in Figure 2.

In the implementation of the computer vision system for target position estimation, we utilized the Intel RealSense D435i camera to acquire image data. The RGB and Depth Images captured by the camera were processed by the main controller using OpenCV. YOLO was employed to recognize feature areas in the image frames, serving as the basis for the subsequent estimation of the target center position and control of the robotic arm’s pose.

### 2.2. Data Acquisition

The target detection task can be framed as a supervised classification problem, where the input to the model is a set of visual features and the output is the class label of the target. To train the model, a labeled dataset must be collected and the relevant features for target identification must be identified.

In our experiments, we utilized the Intel RealSense D435i camera as the image acquisition device. A total of 200 archive box images were captured and labeled with their respective four corners under consistent lighting conditions. Additionally, 100 images of space blocks were acquired and labeled with the top and bottom grid blocks. These images were divided into training and testing sets.

In our experiment, the Intel RealSense D435i camera was used to capture images. A dataset of 200 archive box images and 100 space block images was collected and labeled. To increase the generalization ability of the network, 50 of the archive box images showed surface reflections. To improve the network’s recognition performance, space block images were captured from different angles.The pictures of the file box and the file shelf lattice opening are shown in (a) and (b) of Figure 3, respectively.

The selected feature areas in the archival box and archival shelf grids are shown in (a) and (b) of Figure 4, respectively. In two distinct operating conditions, two distinctive features were selected for recognition purposes. In the case of the archive boxes, the four corners were identified as the feature recognition areas, and the center of the box was estimated based on the positions of the four corners. In the case of the archive compartments, the top and bottom sections of the archive shelf were selected as the feature recognition areas, and the center of the compartment was estimated by evaluating the positions of the corresponding adjacent compartments.

We utilized the YOLO network, which produces prediction boxes and probabilities for each category using a regression approach. By ranking all detection boxes and associating them with relevant feature regions for each archive box, we determined the target center location. The accuracy of determining the center position of the archival box or compartment opening relies on the correct pairing of multiple feature regions. Therefore, we proposed a custom-designed algorithm based on feature region matching for target position estimation.

## 3. Description of the Target Location Estimation Method

### 3.1. Feature Region Ranking Matching

We designed an additional feature region sorting and matching algorithm to address the issue of incomplete feature recognition. The algorithm aims to mitigate the impact of missing feature regions on the robot system’s ability to perform tasks. In this section, we present the implementation of the algorithm using a real-world image of the archival shelf.

As depicted in Figure 5, to ensure accurate determination of the target central position, we addressed the issue of multiple feature regions appearing in the camera’s field of view as the robotic arm moves and the camera angle changes. Our analysis of test images revealed that the feature region facing the camera directly has a higher probability of accurate recognition. To exploit this observation, we developed a sorting and matching algorithm to identify and discard feature regions that exceed a predefined confidence threshold. Our algorithm generates a feature detection frame with each image frame being processed through the YOLO network. We can determine the positions of the four corners of the detection frame to estimate the center of the detection frame (Cn). Then, we use a ranking and matching procedure to identify the most accurate feature regions.
Determination of the position of the YOLO-identified detection frame.We extend the left and right sides of each upper detection box downward and add a settable adjustment variable γ (as marked in Figure 5) for enhancing the system’s immunity to interference in real situations, rowing into a matching region, as shown by the green line in the figure.Determining the center of the lower detection box within the matching area.Determining the inclusion of lower detection boxes within the matching area.Discarding detection boxes that generate matching regions with no corresponding objects.When multiple detection boxes exist in the matching area, the robotic arm adjusts its orientation for improved detection or reports an error.

The process of sorting and matching feature areas for archive boxes is similar to that for archive shelf grids, with the exception of adjusting the variable γ to fit the unique scenario of archive boxes. The system can be adapted to match the specific feature areas of the archive boxes.

### 3.2. Center Location Estimation

The method for recognizing the center of an archive box or shelf opening in a vision system involves the indirect calculation of the center through the identification of local features. The system calculates the center position of the matched feature region, which eliminates the risk of incorrect values influencing the system’s position command to the robotic arm and resulting in significant shaking. This approach ensures the reliability and accuracy of the system’s performance.

In the central location estimation process, feature regions are matched and merged to determine the target’s location information in the image. However, practical issues such as lighting, occlusion, insufficient target structure, the false recognition of similar shapes, etc., can result in inaccurate representation of the target location information by the recognized feature regions.

As shown in Figure 6, two different operational processes of the system for storing or grasping archival boxes are depicted. Due to the differences in the identification of the target and the number of feature regions of the target, two different center position estimation processes are designed for the robotic system to perform the archival storage and grasping tasks, respectively.

In performing the archive box grasping task, as shown in Figure 6a.

Step 1: Loading the image.

Step 2: Identifying the four corner feature regions of the archive box in the picture using YOLOV5, calling the feature region matching module to match each feature region of the archive box, and deleting the regions that fail to be matched.

Step 3: Judging whether the feature regions that have been matched meet the feature region distribution characteristics of the archive box itself.

Step 4: Estimating the center position of the archive box for the frames that meet the feature area distribution of the archive box, discarding the frames that do not meet the feature distribution, and reporting the error.

Step 5: Determining whether it is necessary to perform target center location estimation; skip to Step 1 or end.

When performing the archive storage task, since the robot system is facing the space opening of the archive shelf, there is a slight gap in matching the feature area and judging whether the recognition result satisfies the condition of conducting the center position calculation, as shown in Figure 6b.

Step 1: Loading images. Step 2: Recognizing the checkered feature areas in the picture using YOLOV5, calling the feature area matching module to match the upper and lower checkered feature areas of the archive shelf, and deleting the areas that fail to be matched.

Step 3: Judging whether the feature areas that have been matched meet the distribution characteristics of the archival shelf space block feature areas.

Step 4: Estimating the center position of the archival shelf grid opening for the frames that meet the distribution characteristics of the space opening feature area, discarding the frames that do not meet the feature distribution, and reporting the error.

Step 5: Judging whether it is necessary to carry out target center position estimation; skip to Step 1 or end.

To mitigate errors in the calculated estimated centers caused by missed or false recognition, an automatic feature region matching mechanism is introduced in the object recognition process. This mechanism discards unmatched or out-of-distance feature regions and performs center estimation on well-matched regions. As the vision system operates on a continuous video stream, discarded feature regions can be recalculated in subsequent frames until the center estimation process is stopped.

### 3.3. Access Attitude Control

In this study, we employed an AUBO robotic arm in an engineering application as an actuator for retrieving archive boxes. The central position estimation module provides the position coordinates in the operating space, which are utilized by the robot arm to perform a kinematic inverse operation. This operation calculates the transformation coordinates in the joint space, enabling for the precise control of the end-actuator.

#### 3.3.1. Kinematic Analysis

In conformity with common practices in the field of industrial robotics, the position of the wrist reference point in an AUBO robot arm is determined by the first three joints. On the other hand, the orientation of the wrist is determined by the second three joints. The axes of these latter three joints converge at a single point, which is conventionally used as the reference point for analyzing the dynamics of the arm.

The control system of the robotic arm requires the conversion of the operating space coordinates into joint angle information before it can be utilized for controlling the arm. In the current robotic arm system, the kinematic inverse solution of the position information obtained from the central position estimation module is a prerequisite for calculating the target joint angles. This calculation is necessary to ensure proper control of the robotic arm for the task at hand.

Thanks to the application of the D-H expression, we were able to express the kinematic model of the robotic arm by using a uniform approach in the field of robotics. We define the line perpendicular to the direction of rotation of the joint as the Z-axis, and a line perpendicular to and intersecting the linkage *i* − 1, linkage *i*, and the Z-axis as the X-axis.

We can subtransform from coordinate system (*i* − 1) to coordinate system (*i*) in the following four steps:

Step 1: Rotation around the X-axis of the coordinate system (*i* − 1).

Step 2: Translation along the X-axis of the coordinate system (*i* − 1).

Step 3: Translation of d around the Z-axis.

Step 4: Rotation of θ along the Z-axis.

Based on the robot arm dimension diagram, we build the D-H table, as shown in Table 1.

Based on the joints and links markings in Figure 7, the robotic arm D-H parameters in Table 1 can be obtained. Having acquired the relevant parameters, the calculation of the joint angles of the six axes of the robot arm, which facilitate its movement from the starting position to the end position, can be performed. This constitutes the inverse kinematic solution process for the archive box gripping system, and the specific steps involved are outlined below.

Step 1: To simplify the calculation process of positional data in our system, we utilize a four-dimensional representation through a homogeneous transformation matrix. This matrix converts the 3D coordinates and Euler angles (*z*-*y*-*x*) into a more convenient form for calculations. The reference system for these calculations is the robot base’s coordinate system.

It is known that pos = (*x*, *y*, *z*) euler = (*ℵ*, β, γ)

The first step in this process involves transforming the Euler angles into a three-dimensional rotation matrix. The specific procedure for doing so is as follows.
(1)$$R=\left[\begin{array}{ccc} cℵ & -sℵ & 0\\ sℵ & cℵ & 0\\ 0 & 0 & 1 \end{array}\right]\left[\begin{array}{ccc} c\beta & 0 & s\beta \\ 0 & 1 & 0\\ -s\beta & 0 & c\beta \end{array}\right]\left[\begin{array}{ccc} 1 & 0 & 0\\ 0 & c\gamma & -s\gamma \\ 0 & s\gamma & c\gamma \end{array}\right]=\left[\begin{array}{ccc} cℵc\beta & cℵs\beta s\gamma -sℵc\gamma & cℵs\beta c\gamma +sℵs\gamma \\ sℵc\beta & sℵs\beta s\gamma +cℵc\gamma & sℵs\beta c\gamma -cℵs\gamma \\ -s\beta & c\beta s\gamma & c\beta c\gamma \end{array}\right]$$

The cosine and sine functions are denoted using “*c*” and “*s*”, respectively. Subsequently, *R* and pos are constructed as a four-dimensional homogeneous transformation matrix containing both position and pose information.

This matrix representation facilitates the subsequent calculations required for inverse kinematic solutions in the robotic arm control system.
(2)$$T=\left[\begin{array}{cccc} R_{11} & R_{12} & R_{13} & x\\ R_{21} & R_{22} & R_{23} & y\\ R_{31} & R_{32} & R_{33} & z\\ 0 & 0 & 0 & 1 \end{array}\right]$$

Step 2: According to the formula in step 1, all known poses are converted into a matrix representation to obtain. In order to use AUBO’s inverse solution interface, it is necessary to calculate the pose of end under base when the camera is in the target pose, i.e., the target pose of end. According to the following equation,
(3)$$T_{end\_in\_base\_target}T_{camera\_in\_end}=T_{end\_in\_base}T_{camera\_target\_in\_camera}$$

After transformation, we obtain: (4)$$T_{end\_in\_base\_target}=T_{end\_in\_base}T_{camera\_in\_end}T_{camera\_target\_in\_camera}T_{camera\_in\_end}^{-1}$$

Step 3: In order to use the AUBO interface, it is also necessary to decompose into a position and quaternion expression.
(5)$$T_{end\_in\_base\_target}=\left[\begin{array}{cccc} R_{11} & R_{12} & R_{13} & x\\ R_{21} & R_{22} & R_{23} & y\\ R_{31} & R_{32} & R_{33} & z\\ 0 & 0 & 0 & 1 \end{array}\right]$$

We consider the conversion of a three-dimensional rotation matrix *R* to a quaternion representation. The position vector (*x*, *y*, *z*) is utilized to derive the matrix *R*, and four mathematical formulas are used for the conversion.
(6)$$w=\frac{\sqrt{R_{11}+R_{22}+R_{33}+1}}{2}x=\frac{R_{23}-R_{32}}{4w}y=\frac{R_{31}-R_{13}}{4w}z=\frac{R_{12}-R_{21}}{4w}$$
(7)$$x=\frac{\sqrt{R_{11}-R_{22}-R_{33}+1}}{2}w=\frac{R_{23}-R_{32}}{4x}y=\frac{R_{12}+R_{21}}{4x}z=\frac{R_{31}+R_{13}}{4x}$$
(8)$$y=\frac{\sqrt{-R_{11}+R_{22}-R_{33}+1}}{2}w=\frac{R_{31}-R_{13}}{4y}x=\frac{R_{12}+R_{21}}{4y}z=\frac{R_{23}+R_{32}}{4y}$$
(9)$$z=\frac{\sqrt{-R_{11}+R_{22}+R_{33}+1}}{2}w=\frac{R_{12}-R_{21}}{4z}x=\frac{R_{31}+R_{13}}{4z}z=\frac{R_{23}+R_{32}}{4z}$$

The given system of equations enables the calculation of the corresponding quaternion and the representation of rotational transformations in quaternion form.

Step 4: The given procedure involves utilizing the AUBO interface to input the current six joint angles and the target state of the end-actuator in terms of its position and orientation, represented as a quaternion. By performing inverse kinematic calculations, the corresponding six joint angles required to reach the desired state are determined.

This method allows for the real-time calculation and transmission of the six joint angles to the robot arm control system interface, enabling precise positioning and orientation of the end-actuator.

#### 3.3.2. Design of Storage and Crawling Tasks

The adaptive vision servo system is designed to enhance the safety of archive storage and gripping tasks in the presence of structural variability. The target center position, obtained from the center position estimation system, undergoes filtering and coordinate transformation before being transmitted to the robot arm actuation control system. The result is conversion into joint angle parameters that drive the robot arm to perform the desired actions.

As Figure 8 shows. The design implements a closed-loop control system. The target detection system calculates the center position of the desired target, which is then transformed and passed on to the robotic arm control system for movement control. The target detection system continuously re-estimates the target center position until the difference between the transformed center position and the current robotic arm position falls within a set threshold. When this occurs, the robot arm is deemed to have reached the target position for the task.

## 4. Experimental Results and Analysis

To address the requirement for an unmanned archive repository and to evaluate the viability and stability of the method, we have developed a mobile robot platform featuring autonomous navigation and task planning capabilities. This platform enables us to conduct practical evaluations of the algorithm efficacy and system stability.

### 4.1. Sample Training Experiments

Before evaluating the performance of the robotic system, a training experiment was conducted using a GPU (RTX3060) on a personal computer to train the sample data. The parameters of the main equipment are shown in Table 2.

In this study, self-annotated images of archive boxes, and the top and bottom grids of archive shelves were used as datasets for training, validation, and testing of the proposed algorithm.

Our training experiment yielded highly accurate results for recognizing archive box feature regions. As shown in Figure 9, when the F1 value reached 0.98, as shown in Figure 9a, the confidence level was 0.581, and the accuracy of recognizing all five targets was 1 when the confidence level exceeded 0.833, as shown in Figure 9b. The mAP was 0.991 when the IoU was set to 0.5, as shown in Figure 9c, and the model’s recall was 1 when the confidence level was 0, as shown in Figure 9d. These results indicate the algorithm’s robustness and effectiveness in accurately identifying archive box feature regions.

The performance of the recognition model for the upper and lower shelf feature areas was evaluated through the training results. As shown in Figure 10, the the F1 score reaches a maximum value of 0.89 at a confidence level of 0.819, as illustrated inFigure 10a. The recognition accuracies of the upper and lower grid images, as shown in Figure 10b, are close to 100% when the confidence level is greater than 0.924. The mean average precision (mAP), illustrated in Figure 10c, reaches 0.946 when the intersection over union (IoU) is set to 0.5. Finally, in Figure 10d, the recall of the model reaches 0.98 when the confidence level is 0, indicating its ability to accurately identify the presence of archive boxes in the images.

### 4.2. Center Location Estimation

The Adaptive Vision Servo System is a crucial component of the mobile robot platform and operates within an industrial personal computer (IPC) located within the robot. The detailed configuration of the system is shown in Table 3.

The training experiments were conducted using the RTX3060 GPU on a PC.

Additionally, the prototype for center location estimation is shown in Figure 11. The kinematic parameters of the experimental robotic arm have been given in Figure 7. The experimental environment of the archive shelf is shown in Figure 12.

The proposed algorithm for the central location estimations of archive boxes has demonstrated impressive results in the conducted experiments. This includes, in Figure 13, visualizing the recognized archive boxes through red lines and highlighting their calculated center with a red dot, providing greater interpretability and facilitating the debugging process.

The detection of empty grids was conducted by maneuvering the robotic arm to bring the grids into the camera’s field of view. However, occlusions and viewing angle challenges were encountered, resulting in the incomplete detection of upper and lower grids. A sorting and pairing operation was performed to accurately determine the center position of the space opening.

In Figure 14, the center of the estimated space opening is depicted by a cross with a unique color, which is determined by the feature regions identified by the computer. The performances of the feature region ranking and matching algorithms were evaluated by conducting experiments using samples from the test set to assess the accuracy of the center location estimation of the target.

Experiment based on the flow chart in Figure 15. The results of our experiments indicate that the utilization of the feature region sorting and matching algorithm in the central position estimation method does not impact the number of recognition frames generated by YOLO. To assess the effectiveness of the feature region sorting and matching algorithm, we conducted a comparison between two separate central position estimation modules: one with the feature region sorting and matching algorithm, and one without. The results of this comparison showed that the centrality coordinates output by the two methods were inconsistent in 1.27% of the cases.

The results of the experiments show that the system is capable of estimating the center position of a target from the recognition frame generated by YOLO. The use of a feature region sorting and matching algorithm effectively reduces redundant recognition frames and enhances the accuracy of center position calculation in cases of limited viewing angles.

### 4.3. Analysis of Storage and Gripping Movements

The proposed visual servo system, comprising a camera and a robotic arm, was implemented to perform the task of grasping and storing archives. The target center position estimation was integrated as an input to the robotic arm control system, and transformed into joint actions for precise positioning. The results demonstrate the effectiveness of this approach in accurately completing the storage and gripping processes.

The target center position, as estimated by the system, serves as the input to the robotic arm control system. The goal of the control system is to drive the robotic arm to the specified position, with the end position of the robotic arm serving as the output. The accuracy of the arm’s final execution is crucial in determining the overall performance of the system.

The robotic arm employed in the system is a commercially available device with demonstrated accuracy in its control capabilities. The end position and posture of the arm are determined by inputting the target position into the arm’s control system, which makes use of encapsulated functions to execute the desired motion. The performance of the system is contingent upon the accuracy of the robotic arm’s motion. The robot arm control task is executed through closed-loop feedback control, resulting in the robotic arm reaching the correct position and posture to complete the task.

This logical sequence of operations is critical for the successful completion of the task. The visual servo system first moves the camera and robotic arm to the target position in a specified sequence. Next, the system adjusts the robotic arm’s pose to bring the target into view. Then, the system estimates the target center position and calibrates the robotic arm’s position and orientation. Finally, the robotic arm executes the task with the aid of closed-loop feedback control. This series of steps ensures the precision and accuracy of the robot’s movements, enabling it to complete the task as intended.

During task execution, the robot follows a sequential process that involves moving to the designated position, adjusting the arm’s position to ensure the target is within the field of view, estimating the center position of the target, calibrating the arm’s position and orientation, and finally performing the task. This logical sequence of operations ensures the successful completion of the task. Figure 16 illustrates this sequence of operations. In Figure 16a, the initial state depicts the robot positioned in front of the archive shelf. Figure 16b showcases the state of center position estimation, and feedback control of the robot arm. Figure 16c represents the state of archive box grasping, and Figure 16d demonstrates the state of archive box placement. These four figures collectively illustrate the comprehensive process of the robot system when performing the archive box grasping task.

To evaluate the performance of the proposed system, a comparison was made between a traditional archive box access robot and the proposed system equipped with the vision feedback system. The experimental setup consisted of an archive shelf with 6 levels, each level containing 20 compartments. A total of 80 grasping and storage tasks were carried out for each shelf, with 2 grasping and 2 storage tests being performed for each compartment. The task completion time and success rate were recorded and analyzed.

As illustrated in Table 4, the average completion time for the archive management robot system is slightly higher than the manual completion time for the task. However, in terms of overall efficiency and success rate, the archive management robot system outperforms the manual approach.

## 5. Conclusions

Our study has introduced an adaptive robotic arm gripping system that employs visual feedback and motion control for stable and reliable archive storage in unmanned environments. The system includes a vision system and a robotic arm control system, which function in a feedback loop to estimate the target center accurately and to adjust the arm’s position and posture accordingly. Using the YOLO algorithm for target center position estimation, the proposed system has demonstrated superior performance compared to conventional archive storage systems lacking visual feedback. The experiments have validated the system’s efficacy, making it an invaluable solution for unmanned archive storage rooms. Future work involves expanding the system’s capabilities by adding more training samples for different archive box types and shelves.

## Figures and Tables

**Figure 1 sensors-23-05343-f001:**
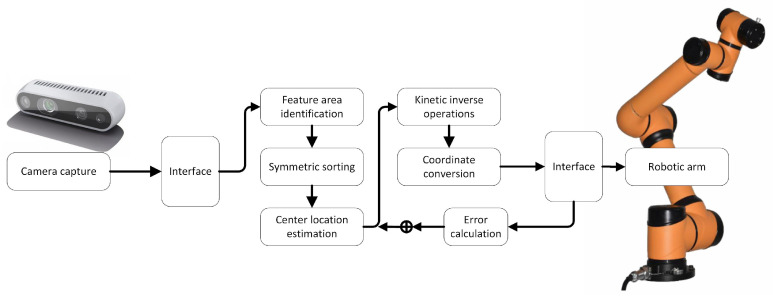
Block diagram of the vision servo system.

**Figure 2 sensors-23-05343-f002:**
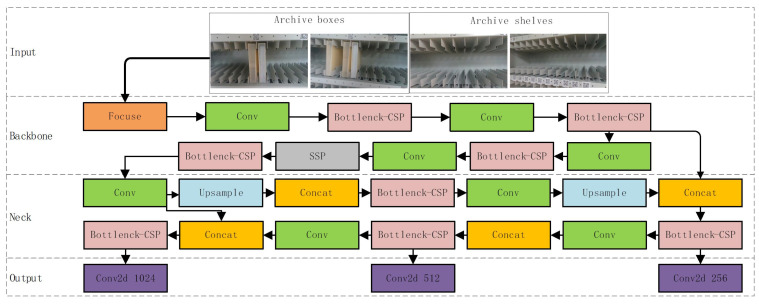
Schematic diagram of the YOLO feature area recognition algorithm.

**Figure 3 sensors-23-05343-f003:**
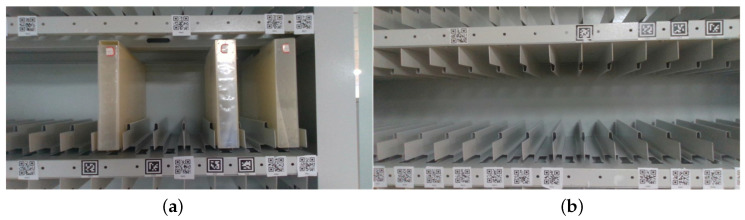
The image in (**a**) displays the paper archive box, while (**b**) showcases the grid aperture of the archive shelf.

**Figure 4 sensors-23-05343-f004:**
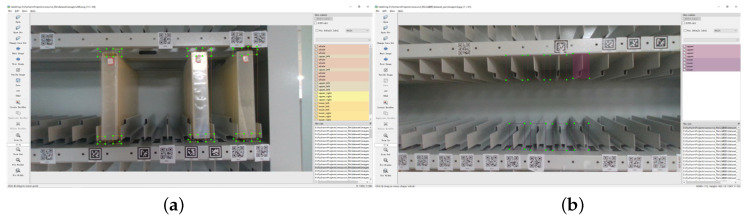
The image in (**a**) displays the feature area of the archive box, while (**b**) shows the feature area of the archive shelf.

**Figure 5 sensors-23-05343-f005:**
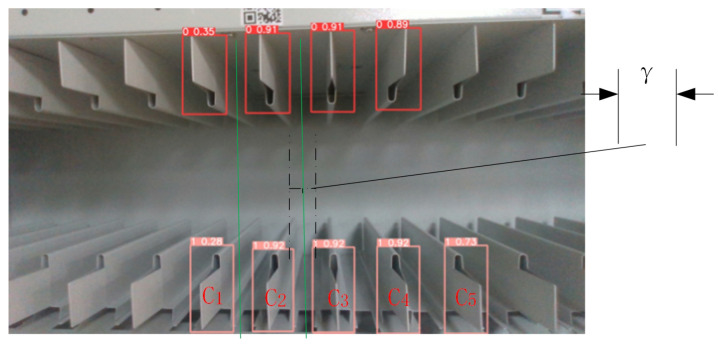
Implementation of the feature area sorting and matching algorithm in the archive shelf.

**Figure 6 sensors-23-05343-f006:**
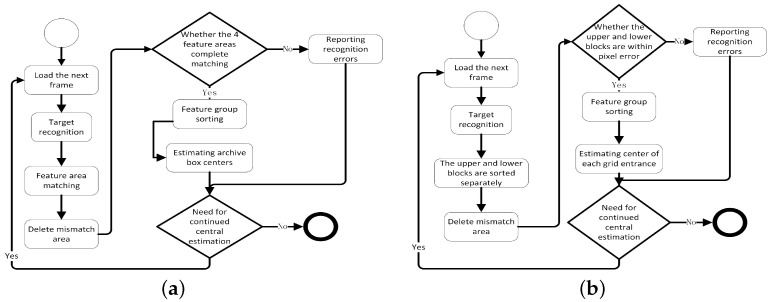
The image in (**a**) shows the flowchart for estimating the center of the archive box, while (**b**) shows the flowchart for estimating the center of the archival shelf space opening.

**Figure 7 sensors-23-05343-f007:**
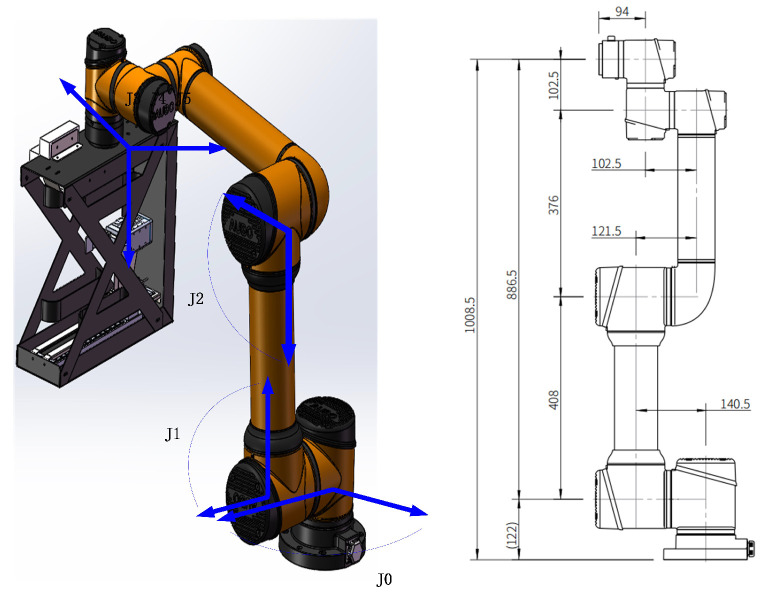
Direction of rotation and dimensional parameters of the robot arm.

**Figure 8 sensors-23-05343-f008:**
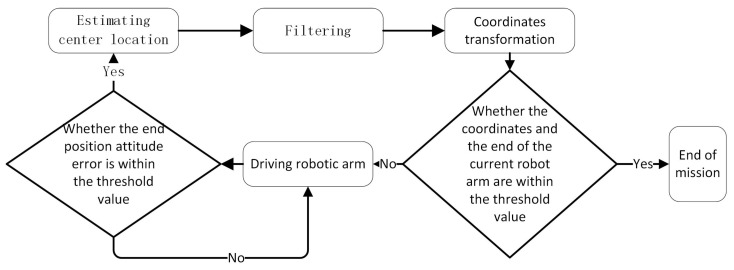
Flow chart of the file box storage and capture system.

**Figure 9 sensors-23-05343-f009:**
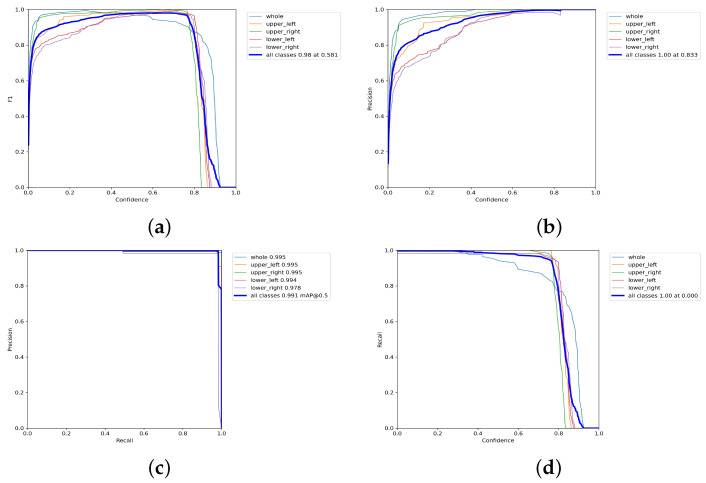
The recognition accuracy of the archive box feature regions was evaluated through the training results. An assessment was carried out on the overall archive box as well as the four individual regions, namely the upper left, upper right, lower left, and lower right. Where (**a**–**d**) represent the reconciled averages of precision and recall; accuracy of prediction; average precision and the relationship between recall and confidence in the archival box recognition training process, respectively.

**Figure 10 sensors-23-05343-f010:**
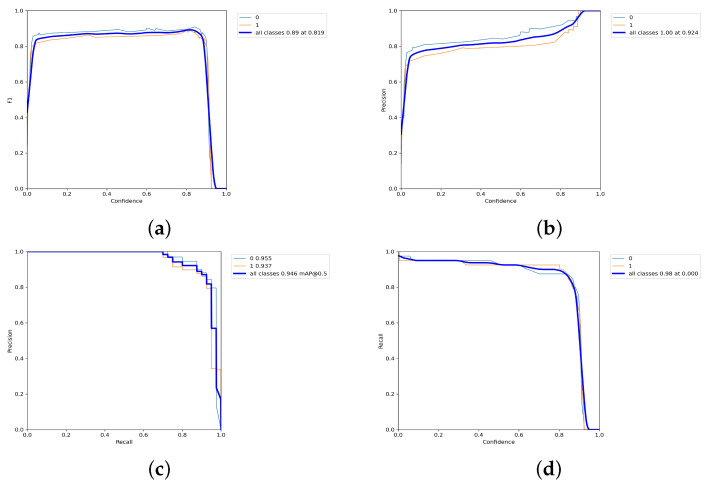
The training results of the upper and lower shelf recognition were evaluated based on a binary classification, with 0 representing the upper shelf and 1 representing the lower shelf. The performance metrics were used to assess the accuracy of the recognition. Where, (**a**–**d**) represent the reconciled averages of precision and recall; predicted accuracy; average precision and the relationship between recall and confidence in the training process of archival shelf upper and lower grid recognition, respectively.

**Figure 11 sensors-23-05343-f011:**
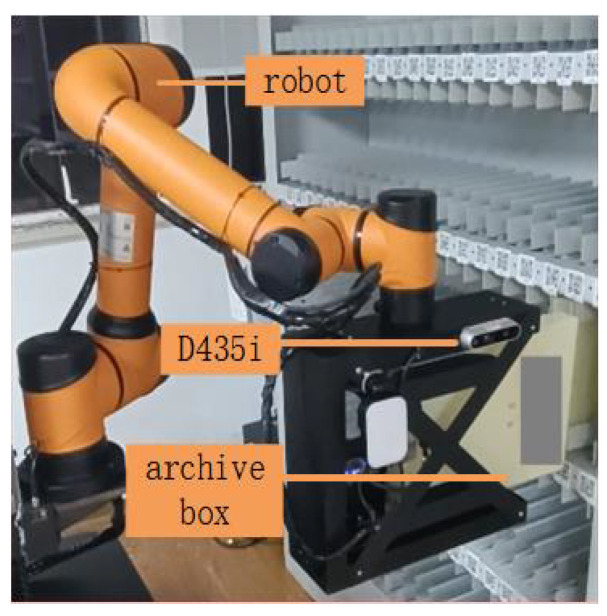
Prototype for center location estimation.

**Figure 12 sensors-23-05343-f012:**
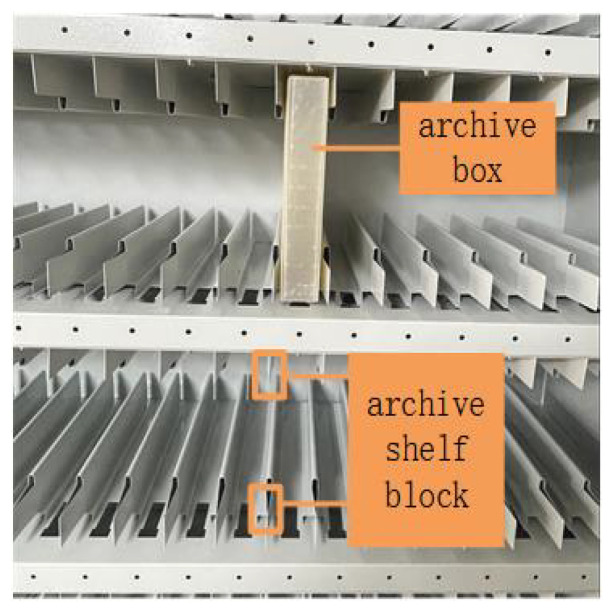
Experimental environment of archive shelf.

**Figure 13 sensors-23-05343-f013:**
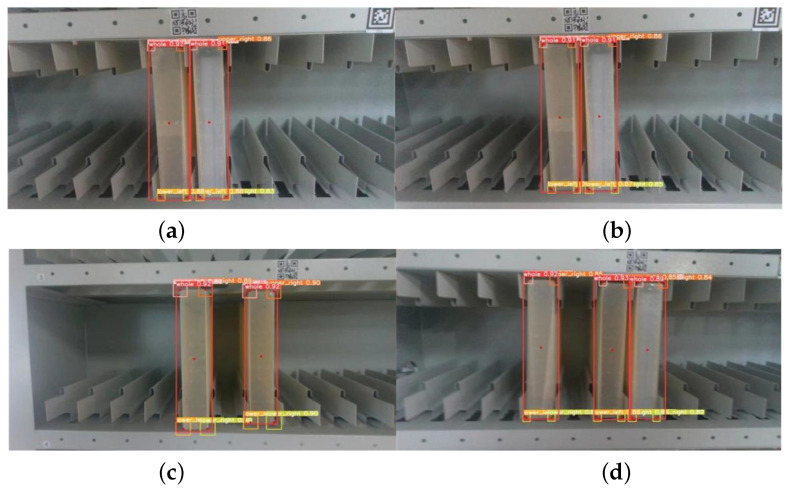
Experiment on the estimation of the center position of archival boxes. The image in (**a**) shows the identification of the viewpoint facing two immediately adjacent archival boxes, (**b**) shows the identification of the viewpoint diagonally facing two immediately adjacent archival boxes, (**c**) shows the identification of the viewpoint facing two separated archival boxes, and (**d**) shows the identification of the viewpoint diagonally facing two separated archival boxes.

**Figure 14 sensors-23-05343-f014:**
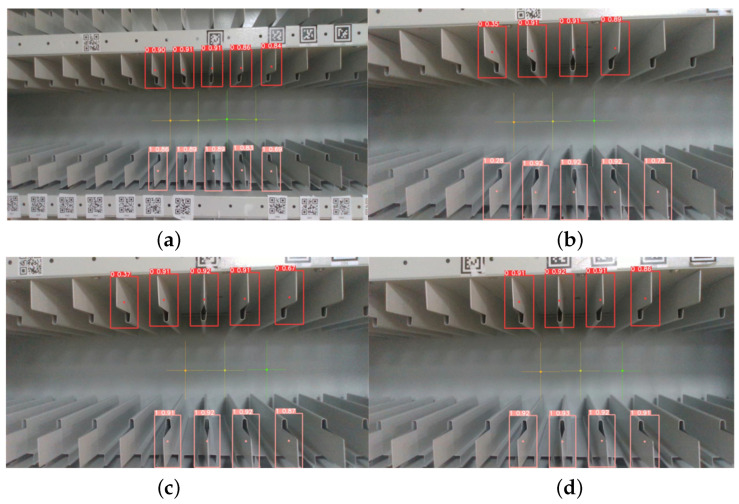
Experiment on the estimation of the center position of the archival shelf space opening. The image in (**a**) shows the case where the viewpoint is oblique to the archival shelf and produces a symmetrical top and bottom identification frame, (**b**) shows the case where the viewpoint is facing the archival shelf and produces an asymmetrical top and bottom identification frame, (**c**) shows the case where the viewpoint is oblique to the archival shelf and produces an asymmetrical top and bottom identification frame, and (**d**) shows the case where the viewpoint is facing the archival shelf and produces a symmetrical top and bottom identification frame.

**Figure 15 sensors-23-05343-f015:**
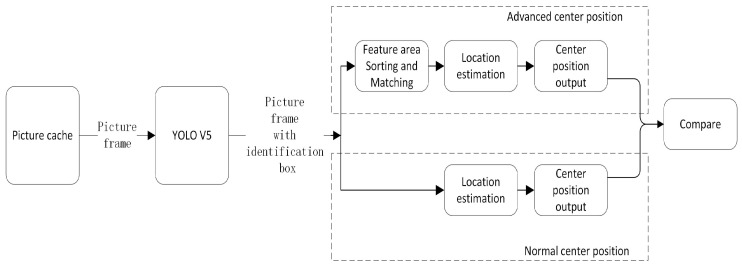
Block diagram of the comparison of experimental flow, for whether to add the feature region sorting matching algorithm.

**Figure 16 sensors-23-05343-f016:**
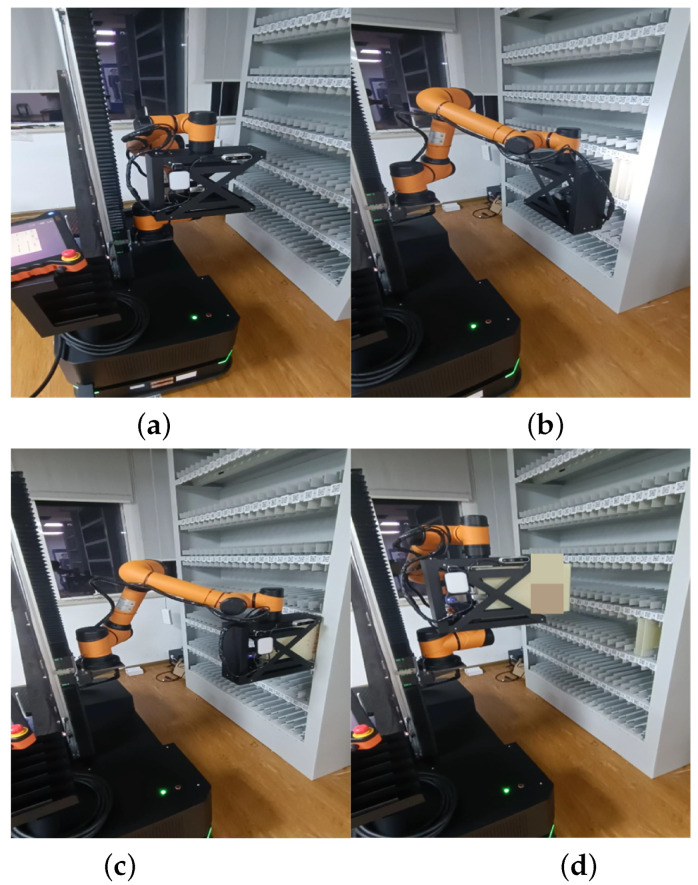
Pictures of the robotic arm performing the task process; where (**a**) is a picture of the robotic arm starting, (**b**) is a picture of the target calibration, (**c**) is a picture of the target grasping, and (**d**) is a picture of the target placement.

**Table 1 sensors-23-05343-t001:** D-H table of the experimental robotic arm.

${\mathrm{Link}}_{i}$	$a_{i}$ (mm)	${ℵ}_{i}{(}^{\circ })$	$d_{i}$ (mm)	$\theta_{i}{(}^{\circ })$
1	0	0	98.5	$\theta_{1}$
2	0	$-\frac{\pi }{2}$	121.5	$\theta_{2}$
3	408	π	0	$\theta_{3}$
4	376	π	0	$\theta_{4}$
5	0	$-\frac{\pi }{2}$	102.5	$\theta_{5}$
6	0	$\frac{\pi }{2}$	94	$\theta_{6}$

**Table 2 sensors-23-05343-t002:** Data training experiment environment.

OS	Windows 10
Programming environment	Python 3.10
CUDA	Version 10.2
Hardware platform	CPU (i5-12600K), Memory (16 GB), GPU (RTX3060).
Object detector	YOLOV5
Input source	Intel RealSense D435i

**Table 3 sensors-23-05343-t003:** Robotic system experimental environment.

OS	Ubuntu 20.04
Programming environment	Python 3.10
CUDA	Version 10.2
Hardware platform	CPU (i5-4200U), Memory (8 GB).
Object detector	YOLOV5
Input source	Intel RealSense D435i

**Table 4 sensors-23-05343-t004:** Comparison of manual task execution and machine task execution times.

Number of File Shelves	Average Manual Execution Time (s)	Adaptive File Access System Average Time to Perform a Task (s)	Average Execution Time for Robot Complete Manufacturers (s)	Task Success Rate of Adaptive System Execution	Task Success Rate of Robot Complete Manufacturers
1	2.41	2.74	2.97	1	0.975
2	2.15	2.71	2.99	1	0.975
3	2.41	2.44	2.46	1	1
4	1.94	2.51	2.71	1	1
5	2.33	3.41	3.84	1	0.9875
6	2.65	3.34	3.84	1	0.925

## Data Availability

Data available upon request from the authors. The data that support the findings of this study are available from the corresponding author, upon reasonable request.

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
