# Peer review of "Target Detection-Based Control Method for Archive Management Robot"

_sensors, 2023, doi:10.3390/s23115343_

Round 1
Reviewer 1 Report
This investigation has been proposed 3 a paper archive access system with adaptive recognition for handling complex archive box access 4 scenarios. Their proposed system comprises a vision component that employs the YOLO V5 algorithm to identify 5 feature regions, sort and filter data, and estimate the target center position, as well as a servo control 6 component. This study proposes a servo-controlled robotic arm system with adaptive recognition 7 for efficient paper-based archive management in unmanned archives.
Finally, their study has introduced an adaptive robotic arm gripping system that employs 398 visual feedback and motion control for stable and reliable archive storage in unmanned 399 environments.
The paper should be improved with broad discussion relating the performance of the proposed algorithm rather than conventional algorithms. Joint angles have been shown on Figure 16, should be indicated by using a table. What are the signal parameters of the block diagram on Figure 15. On Figure 13-14, all pictures a)-d) should be described in details. Dynamic and Kinematic parameters of the industrial robot should be given in table as shown on Figure 11. Figure 10 should be outlined with the description of a)-d). It is not clear to understand the meaning of descriptions.
"The performance of the recognition model for the upper and lower shelf feature areas was 314 evaluated through the training results.The F1 score reaches a maximum value of 0.89 at a 315-confidence level of 0.819, as illustrated in Figure a. The recognition accuracy of the upper 316 and lower grid images, as shown in Figure b, is close to 100% when the confidence level 317 is greater than 0.924. The mean average precision (mAP), illustrated in "Figure c, (??) reaches 318 0.946 when the intersection over union (IoU) is set to 0.5. Finally, in Figure d, the recall of 319 the model reaches 0.98 when the confidence level is 0, indicating its ability to accurately 320 identify the presence of archive boxes in the images"
The previous description does not match the figures. This text should be re-checked and write again.
What are the main differences between 2 flowcharts on figure 6? , both should be outlined in the text.
Paper is related to reference 15. "Intisar, M.; Khan, M.M.; Islam, M.R.; Masud, M. Computer Vision Based Robotic Arm Controlled Using Interactive GUI. Intelligent 455 Automation & Soft Computing 2021, 27"
Some text description should be improved with technical English and broad description on Discussion.
Author Response
Response to Reviewer 1 Comments
Dear reviewers, based on your valuable suggestions, I have divided the comments into the following 8 points, and will reply and revise them one by one.
Point 1:
The paper should be improved with broad discussion relating the performance of the proposed algorithm rather than conventional algorithms.
Response 1:
We have added pictures of the actual control of the robot arm in the process of modification, in order to be able to describe the robot's movement more visually.
Point 2:
Joint angles have been shown on Figure 16, should be indicated by using a table.
Response 2:
The graph of the change of joint angles was intended to reflect, during the execution of tasks by the robotic system, the change of the robotic arm. Subsequent consideration of the use of the change of joint angles does not visually describe the change of the robotic arm. At the same time, the change of the joint angle of the robotic arm is not important in this work; the main focus is on the starting pose of the robotic arm and the ending pose of each task. Therefore, we replaced the joint angle diagram of the robotic arm with the physical diagram of the pose in the experiment to demonstrate the completeness of the experiment.
Point 3:
What are the signal parameters of the block diagram on Figure 15.
Response 3:
In order to better express the meaning of the experiment in Figure 15, we replaced the flow chart and modified the title of the figure.
To better convey the meaning of the experiment in Figure 15, we replaced the flow chart and modified the name of the diagram.
Replace the original figure name with the following:”Block diagram of the comparison experimental flow, for whether to add the feature region sorting matching algorithm.”
Point 4:
On Figure 13-14, all pictures a)-d) should be described in details.
Response 4:
We have added the following note in the body of the text:
Figure 13:
Experiment on the estimation of the center position of archival boxes. Fig. (a) shows the identification of the viewpoint facing two immediately adjacent archival boxes, Fig. (b) shows the identification of the viewpoint diagonally facing two immediately adjacent archival boxes, Fig. (c) shows the identification of the viewpoint facing two separated archival boxes, and Fig. (d) shows the identification of the viewpoint diagonally facing two separated archival boxes.
Figure 14:
Experiment on the estimation of the center position of the archival shelf space opening. Fig. (a) shows the case where the viewpoint is oblique to the archival shelf and produces a symmetrical top and bottom identification frame, Fig. (b) shows the case where the viewpoint is facing the archival shelf and produces an asymmetrical top and bottom identification frame, Fig. (c) shows the case where the viewpoint is oblique to the archival shelf and produces an asymmetrical top and bottom identification frame, and Fig. (d) shows the case where the viewpoint is facing the archival shelf and produces a symmetrical top and bottom identification frame.
Point 5:
Dynamic and Kinematic parameters of the industrial robot should be given in table as shown on Figure 11.
Response 5:
Added a note near Figure 11 of the article:The kinematic parameters of the experimental robotic arm have been given in Figure 7.
Point 6:
Figure 10 should be outlined with the description of a)-d). It is not clear to understand the meaning of descriptions.
Response 6:
The description is added to the main text, and the figure number corresponding to the description is added.
Point 7:
"The performance of the recognition model for the upper and lower shelf feature areas was 314 evaluated through the training results.The F1 score reaches a maximum value of 0.89 at a 315-confidence level of 0.819, as illustrated in Figure a. The recognition accuracy of the upper 316 and lower grid images, as shown in Figure b, is close to 100% when the confidence level 317 is greater than 0.924. The mean average precision (mAP), illustrated in "Figure c, (??) reaches 318 0.946 when the intersection over union (IoU) is set to 0.5. Finally, in Figure d, the recall of 319 the model reaches 0.98 when the confidence level is 0, indicating its ability to accurately 320 identify the presence of archive boxes in the images"
The previous description does not match the figures. This text should be re-checked and write again.
Response 7:
The descriptions of the figures in the article were misunderstood by you due to the layout and our lack of attention to the figure numbers corresponding to the text. Therefore, we have added the figure numbers in the text section in the hope that you can access the information more convenient.
Point 8:
What are the main differences between 2 flowcharts on figure 6? , both should be outlined in the text.
Response 8:
We added the following description to the body section near Figure 6 in anticipation of a clearer representation of the robot's workflow while performing the task:
“As shown in Figure 6, two different operational processes of the system for storing or grasping archival boxes are depicted. Due to the differences in the identification of the target and the number of feature regions of the target, two different center position estimation processes are designed for the robotic system to perform the archival storage and grasping tasks, respectively.
In performing the archive box grasping task, as shown in Figure (a)
Step1: loading the image;
Step2: identifying the four corner feature regions of the archive box in the picture by YOLOV5, calling the feature region matching module to match each feature region of the archive box, and deleting the regions that fail to be matched;
Step3: judge whether the feature regions that have been matched meet the feature region distribution characteristics of the archive box itself;
Step4: estimate the center position of the archive box for the frames that meet the feature area distribution of the archive box, discard the frames that do not meet the feature distribution and report the error;
Step5: determine whether it is necessary to perform target center location estimation; skip to Step1 or end.
“
“
When performing the archive storage task, since the robot system is facing the space opening of the archive shelf, there is a slight gap in matching the feature area and judging whether the recognition result satisfies the condition of conducting the center position calculation. As shown in figure (b).
Step1: Loading images;
Step2: recognizing the checkered feature areas in the picture by YOLOV5, calling the feature area matching module to match the upper and lower checkered feature areas of the archive shelf, and deleting the areas that fail to be matched;
Step3: judge whether the feature areas that have been matched meet the distribution characteristics of the archival shelf space block feature areas;
Step4: estimate the center position of the archival shelf grid opening for the frames that meet the distribution characteristics of the space opening feature area, and discard the frames that do not meet the feature distribution and report the error;
Step5: judge whether it is necessary to carry out target center position estimation; skip to Step1 or end.
“
Reviewer 2 Report
The authors propose a YOLO V5-based algorithm for target location estimation and an action execution control method for file management tasks with a robotic arm. Authors are strongly encouraged to highlight their main contribution to the current state of the art, which is not very clear. The points provided in lines 106-113 are not enough to support the novelty. YOLO V5 is a well-known detection method. The kinematic analysis presented in section 2.3.1 is also well known. The novelty seems to be in the practical application, but in this case, it is necessary to explain in more detail the challenges that are being solved and explain the advantages of the proposed method with respect to other baseline studies.
Regarding the control of the robotic arm, a more detailed explanation of the implemented methods is required. For example, it is not clear how the graping is carried out. Is the same orientation of the gripper always used in all cases, or are there variations? If there are variations, how is the best orientation of the gripper estimated?
Figure 16 does not provide relevant information.
Please include the figure reference on line 187.
Minor editing of English language required. Please review typos on lines 178, 257
Author Response
Response to Reviewer 2 Comments
Dear reviewers, based on your valuable suggestions, I have divided the comments into the following 8 points, and will reply and revise them one by one.
Point 1:
The authors propose a YOLO V5-based algorithm for target location estimation and an action execution control method for file management tasks with a robotic arm. Authors are strongly encouraged to highlight their main contribution to the current state of the art, which is not very clear. The points provided in lines 106-113 are not enough to support the novelty. YOLO V5 is a well-known detection method. The kinematic analysis presented in section 2.3.1 is also well known. The novelty seems to be in the practical application, but in this case, it is necessary to explain in more detail the challenges that are being solved and explain the advantages of the proposed method with respect to other baseline studies.
Response 1:
In response to the key issues you raised, we have revised and replaced the original text with:“The utilization of the YOLOV5 method for the recognition of archival boxes and archival shelf grids was explored and its feasibility was experimentally verified through recognition of marked feature areas.”is replaced with“Applying YOLOV5's method to the identification of archival boxes and archival shelf grids, and experimentally verifying the feasibility of identifying the marked feature areas, demonstrating that the method can be applied to the niche area of paper archival access.”
“Experimental results demonstrate that the proposed visual positioning method exhibits improved speed and accuracy compared to the conventional 2D positioning approach.”is replaced with“By comparing with other commercial file management robots on the market, the comparison experiment results show that using the visual positioning method is faster and more accurate than the QR code positioning method, which has better practical value.”
For visual convenience, you can view it directly in the text on line 105-119.
Point 2:
Regarding the control of the robotic arm, a more detailed explanation of the implemented methods is required. For example, it is not clear how the graping is carried out. Is the same orientation of the gripper always used in all cases, or are there variations? If there are variations, how is the best orientation of the gripper estimated?
Response 2:
You and another reviewer jointly raised questions about this joint transformation diagram, and since you raised different points, we have replaced the image here with a physical picture of the robotic arm performing the task in order to describe the integrity of the experiment. This better conveys the specifics of the grasping time.
In order to describe the operation status of the system, the original graph of joint angle change in Fig. 16 is replaced with a realistic graph of the robot working under different conditions in this time.
Point 3:
Figure 16 does not provide relevant information.
Response 3:
In response to your valuable comments, we have added a description of the newly replaced image 16. We replaced the original Figure 16 with a physical picture, which better illustrates the experiment, and modified the description of the picture in the corresponding position to improve the completeness of the experiment.
Point 4:
Please include the figure reference on line 187.
Response 4:
To address this, we modified Figure 5 and added variable labeling, and modified lines 191-197 of the original text.
Round 2
Reviewer 1 Report
Dynamic and kinematic parameters of the robot should be given in table
Some misspelling should be corrected
Author Response
Point 1: Dynamic and kinematic parameters of the robot should be given in table
Response 1:
In response to your suggestion we have added the D-H (Table 1) table of the robotic arm and the corresponding textual description for expressing the kinematic parameters of the robotic arm near line 273 of the original article to improve the completeness of the article.
You can easily find the revised section near line 273 of the article.
Point 2: Some misspelling should be corrected
Response 2:
In response to your suggestion, we have checked the spelling of the article, and we have adopted American English for the article. At the same time, we read through the entire article twice and corrected any ambiguities in the text, so you can clearly see the corrected parts of the article in revision mode.
Reviewer 2 Report
Authors have addressed most of my comments and suggestion. Therefore I recommend the publication of the paper in its present form.
English language fine. No issues detected.
Author Response
Point 1: Authors have addressed most of my comments and suggestion. Therefore I recommend the publication of the paper in its present form.
Response 1:
Thank you very much for your approval, reviewers. Using the second round of revisions, we have re-ranked the English presentation and made some changes, if you are interested you can clearly see the parts we have revised in the revision mode.
Point 2: English language fine. No issues detected.
Response 2:
Thanks again to you, I look forward to sending you a better article soon and look forward to cooperating with you.